# Muscle Cortisol Levels, Expression of Glucocorticoid Receptor and Oxidative Stress Markers in the Teleost Fish *Argyrosomus regius* Exposed to Transport Stress

**DOI:** 10.3390/ani11041160

**Published:** 2021-04-18

**Authors:** Martina Bortoletti, Lisa Maccatrozzo, Giuseppe Radaelli, Stefano Caberlotto, Daniela Bertotto

**Affiliations:** 1Department of Comparative Biomedicine and Food Science (BCA), University of Padova, I-35122 Padova, Italy; martina.bortoletti@phd.unipd.it (M.B.); lisa.maccatrozzo@unipd.it (L.M.); daniela.bertotto@unipd.it (D.B.); 2Valle Cà Zuliani Società Agricola Srl, Rovigo, Via Pila 48, I-45018 Pila di Porto Tolle, Italy; stefano.caberlotto@vallecazuliani.it

**Keywords:** fish transport, cortisol, *gr*, HSP70, oxidative stress markers, meagre, animal welfare

## Abstract

**Simple Summary:**

Over the years, increasing attention has been given to welfare of farmed animals, but with a focus on mammalian species. Nevertheless, even non-mammalian ones deserve to be considered and especially fish. Aquaculture, in fact, is a globally and rapidly growing industry, since a large variety of fish is consumed every day as an essential source of protein and healthy lipids for human nutrition. However, in common aquaculture practices, such as transportation, fish are exposed to a whole range of potentially adverse stimuli which may substantially influence the welfare of the fish. On this basis, the present study aimed to assess the welfare status of meagre juveniles subjected to a 48 h transport. For this purpose, stress response in terms of cortisol levels, glucocorticoid receptor and oxidative stress markers’ expression has been evaluated. Results revealed that fish were stressed during loading on the truck and within 16 h from departure and did not fully recover even at the end of the transport. This work highlights how the procedures prior to transport, in particular, loading, may represent a potential welfare concern rather than transport itself. However, it remains essential to follow live animal commercial transportation directions in order to protect animal health and welfare during transport.

**Abstract:**

Fish commercial transport is an ordinary practice in the aquaculture industry. This study aimed to investigate the effect of a 48 h transport stress on stress response of meagre (*Argyrosomus regius*) juveniles. Radioimmunoassay (RIA) and Real-Time PCR were used to evaluate muscle cortisol levels and to assess glucocorticoid receptor (*gr*) gene expression in fish muscle and liver, respectively. Presence and localization of various oxidative stress markers were investigated in different tissues by immunohistochemistry. A significant increase in muscle cortisol levels was observed after loading but a significant decrease occurred after 16 h from departure even without returning to control levels. Molecular analysis on stress response revealed an increase in muscle *gr* expression after fish loading that started decreasing during the travel returning to the control level at the end of the transport. Instead, no differences in liver *gr* expression were observed along the different sampling points. Immunostaining for heat shock protein 70 (HSP70), 4-hydroxy-2-nonenal (HNE), nitrotyrosine (NT) and 8-hydroxy-2’-deoxyguanosine (8-OHdG) antibodies was detected in several organs. Notably, a higher NT immunostaining intensity was evident in skin and gills of the transported animals with respect to controls. Results demonstrated that cortisol and *gr* are useful indicators of stressful conditions in transported fish.

## 1. Introduction

During the last years, great attention has been directed towards the improvement of farmed animals’ welfare, including fish. Transportation of live fish is a common practice that includes a complex of factors such as handling, air exposure, constraint and low oxygen levels. All these factors are known to induce a stress response in fish, increasing their metabolic rate and overexertion, coupled with a rapid deterioration of water quality. Therefore, given the paramount role that transportation plays in the welfare of farmed fish, several institutions and organizations, both at national (Italian Ministry of Health) and international (European Food Safety Authority and World Animal Health Organization) level, designated specific guidelines providing the best procedures available to ensure fish welfare during transport [1,2,3].

The overall effect of stress is the activation of the hypothalamic-pituitary-interrenal axis (HPI) and the release of catecholamines and glucocorticoids. In teleosts, cortisol is the primary glucocorticoid that is released in response to stressor activation of the hypothalamus-pituitary-interrenal axis. Cortisol induces secondary responses related mainly to energy requirements and it represents one of the most important stress indicators in fish. The target tissue action of this hormone is primarily mediated by the intracellular glucocorticoid receptor (GR), a ligand-bound transcription factor [4]. Among the different organs, liver and muscle are probably the sites of choice for evaluating its gene expression as they are directly and strongly involved in protein catabolism and gluconeogenesis. Amino acids released from the catabolic effects of GR-mediated cortisol signaling on skeletal muscle tissue are mobilized to liver, where they will be used as substrates for hepatic gluconeogenesis [5,6,7,8]. In teleosts, glucocorticoid synthesis takes place in the head kidney interrenal tissue and it is mediated by steroidogenic enzymes [9]. Glucocorticoids play important roles in the homeostasis of many biological systems, including stress response [10,11].

At the cellular level, the formation of reactive oxygen species (ROS) often derives from a stress event [12]. These products are biologically generated during metabolism although in stress conditions, their synthesis is greater than the ability of cells to remove them, leading to lipid peroxidation, protein carbonils’ formation, DNA damage and cell death [13,14]. One of the most important ROS is the superoxide radical, which reacts with nitric oxide giving rise to peroxynitrite, a potent oxidant that may oxidize proteins, lipids and DNA. Nitrotyrosine (NT) is a relatively stable marker for peroxynitrite production [15].

Lipid peroxides are unstable indicators of oxidative stress in cells that decompose to form more complex and reactive compounds such as 4-hydroxy-2-nonenal (HNE), which is a natural by-product of lipid peroxidation. HNE, the most abundant and toxic α, β-unsaturated aldehyde, originates from the β-cleavage of hydroperoxides from ω-6 PUFAs and is mainly involved in the inhibition of protein and DNA synthesis, in the inactivation of enzymes, and is also a potent mutagen agent [16]. 8-hydroxy-2’-deoxyguanosine (8-OHdG), a product of deoxyguanosine oxidation, is an example of DNA damage due to bases’ modification [17]. 8-OHdG level as biomarker of DNA damage has been applied in assessing content of DNA damage as it is a major product of oxidative DNA damage with clear mutagenic potential for G to T transversions [18].

In a previous work, the distribution of inducible Heat Shock Protein 70 (HSP70), which is expressed in response to stressful stimuli, has been detected in different tissues of sea bass (*Dicentrarchus labrax*) subjected to transport stress [19]; moreover, in *Cyprinus carpio*, the inducible form (HSP70) has been localized in the epithelia of renal tubules, gills and skin of animals subjected to transport stress [20].

The aim of the present study was to investigate the muscle cortisol levels, the expression of glucocorticoid receptor in muscle and liver and the cellular distribution of HSP70, HNE, NT and 8-OHdG in several tissues by immunohistochemical approach in the teleost fish meagre (*Argyrosomus regius*) exposed to commercial transport.

## 2. Materials and Methods

All procedures and animal care were in compliance with Council Regulation (EC) no. 1/2005 on the protection of animals during transport and Directive (EC) 86/609/EEC on the protection of animals used for experimental and other scientific purposes.

### 2.1. Travel Condition from Fish Hatchery and Sampling

Fish transport has been carried out from Monfalcone (Gorizia, Italy) to Ajaccio (Corsica) by a commercial truck and lasted 48 h. Tanks (2.7 m^3^) were insulated and provided with aeration and temperature control. Meagre juveniles’ weight was 8 g and density in tanks was kept at 20 kg/m^3^; 72 h fasting period was performed. During the transport, temperature and oxygen levels were respectively 19–23 °C and 16–22 ppm.

Fish were sampled before (control) and after loading that lasted two hours and during and at the end of transport event (16 and 48 h after departure). For each sampling point, 35 fish have been sacrificed with MS222 excess (1 g/L Sandoz, Milan, Italy).

For cortisol measurement, muscle from 20 fish was collected from the caudal peduncle and immediately frozen and kept in dry ice. For biomolecular analysis, muscle and liver samples from 10 fish were collected and immersed in RNA later (Life Technologies, Carlsbad, CA, USA) and stored at 20 °C until analysis. For immunohistochemistry, several organs (gills, oropharyngeal cavity, stomach, intestine, liver, pancreas, lateral muscle, and skin) from 5 fish were fixed in paraformaldehyde in phosphate buffer saline (PBS, 0.1 M, pH 7.4) at 4 °C overnight, PBS washed, dehydrated and embedded in paraffin. Serial sections of 4 μM were obtained using a microtome.

### 2.2. Cortisol Measurement

Cortisol was measured with a microtiter RIA, as described by Bertotto et al. [21], validated for the meagre, after extraction of 100 mg of muscle in diethyl ether. The sensitivity of the assay was 3.125 pg well^−1^.

### 2.3. RNA Extraction and Real-Time PCR

Qualitative reverse transcription/PCR and quantitative Real-Time PCR were performed by following the methods detailed in Bertotto et al. [22]. Total RNA was extracted from muscle and liver dissected from fish at different times during transportation (10 animals for each sampling point): 1. at the fish hatchery, before transport; 2. just loaded on the truck; 3. during transport; 4. at the end of transportation, using TRIZOL reagent (Life Technologies, Carlsbad, CA, USA) following the manufacturer’s protocol.

PCRs were performed in triplicate in a Real-Time PCR 7500 thermal cycler (Applied Biosystems, by Life Technologies). Specific primers for Glucocorticoid receptor gene (forward 5′-GCCTTTTGGCATGTACTCAAACC-3′ and reverse 5′-GGACGACTCTCCATACCTGTTC-3′) and for β-actin (forward 5′-ACCCTGTCCTGCTCACAGAG-3′ and reverse 5′-GGGAGTCCATAACAATACCAGTG-3′) used as reference gene, were designed with Primer Express software version 3.0 (Applied Biosystems, Life Technologies). 

Relative quantification of the expression of glucocorticoid receptor gene was performed using β-actin as the housekeeping genes to standardize results. 

### 2.4. Immunohistochemistry (IHC)

All the antibodies used in this study are reported in Table 1. Immunohistochemical staining was performed using the Elite ABC KIT system (Vector Laboratories, Inc., Burlingame, CA, USA) as described in Pascoli et al. [23]. Briefly, after endogenous peroxidase activity and non-specific binding sites were blocked, sections were incubated with primary antibodies (see Table 1), at 4 °C overnight. After PBS washing, sections were incubated with anti-rabbit or Ig antibodies, biotin-conjugated anti-mouse (Dakocytomation), PBS washed and reacted with peroxidase-labeled avidin-biotin complex (Vector Laboratories, Inc., Burlingame, CA, USA). The immunoreactive sites were visualized using 3.3’-diaminobenzidine tetrahydrochloride (DAB, Sigma, Milan, Italy). To detect structural details, sections were counterstained with Mayer’s haematoxylin. Immunostaining specificity was validated by incubating sections with: (i) PBS instead of the specific primary antibodies (see Table 1); (ii) preimmune sera instead of the primary antisera; (iii) PBS instead of the secondary antibodies. The results of these controls were negative (i.e., staining was abolished).

### 2.5. Statistical Analysis

Statistical analysis was carried out by means of STATISTICA 8.0 (StatSoft, Tulsa, OK, USA). The cortisol concentration values, in the different transport phases, were previously transformed using a logarithmic function and then compared by means of one-way analysis of variance (ANOVA); in the presence of significant differences between the means, a post hoc comparison of Honestly Significant Difference (HSD) for unbalanced designs was carried out.

The analysis of Real-Time PCR data on the presence of the mRNA for the glucocorticoid receptor (GR), both in the muscle and in the liver and in the various stages of transport, was carried out using the GLM (General Linear Model) approach; also, in this case, the data were transformed with a logarithmic function.

## 3. Results

### 3.1. Cortisol Measurement

The cortisol assay showed acceptable parallelism and reproducibility (linear regression curve y = 11.2x + 0.1; regression coefficient R^2^: 0.99; CV% intra-assay = 1.7). The recovery test with value higher than 86% confirmed the efficiency of steroid extraction method. Muscle cortisol increased in fish after loading (*p* < 0.0001) but a significant decrease occurred soon after 16 h from departure even without returning to the control levels (Figure 1).

### 3.2. RNA Expression

Relative quantification of glucocorticoid receptor gene reveals an increase in muscle after fish loading (*p* < 0.001) that has been maintained after 16 h from departure but decreased returning to the control level at the end of the transport (Figure 2). No differences in expression of glucocorticoid receptor in the different sampling points were detected in the liver (*p* = 0.4).

### 3.3. HSP70 Immunohistochemistry

Immunoreactivity to HSP70 antibody was detected in several tissues and organs although no differences were found between control and stressed animals (Table 2).

In particular, at cellular level, immunoreactivity to HSP70 antibody was detected in: (i) hepatocytes of liver parenchyma (Figure 3A, insert in A); (ii) cells lining the gastric glands of the stomach (Figure 3B); (iii) cells of gill epithelium at the level of filaments and lamellae (Figure 3C); (iv) the muscle fibre of skeletal lateral muscle (Figure 3D). Immunoreactivity was also present in the epithelia of skin and intestine (Table 2).

### 3.4. HNE Immunohistochemistry

Positive immunostaining of the anti-HNE antibody was exhibited in the oropharyngeal cavity, stomach, liver, intestine, and skin, although no differences were detectable between control and stressed animals (Table 2). In the oropharyngeal cavity, HNE-immunostaining was visible in the cytoplasm of the epithelial cells of the mucosa (Figure 4A). In the stomach, the cytoplasm of cells lining the gastric glands exhibited a moderate HNE-immunostaining (Figure 4B, insert in B). In the liver, immunostaining was diffusely detectable in the cytoplasm of hepatocytes (Figure 4C). In the intestine, an evident HNE-immunostaining was visible in the cytoplasm of enterocytes (Figure 4D).

### 3.5. NT and 8-OHdG Immunohistochemistry

The anti-NT antibody exhibited positivity (Table 2) in the epithelial cells of skin (Figure 5A), in the cytoplasm of hepatocytes, as well as in the pancreatic glands (Figure 5B). Moreover, immunopositivity was evident in the enterocytes of the intestinal mucosa (Figure 5C) and in the epithelial cells lining both filaments and lamellae of gills (Figure 5D). The immunostaining intensity observed in skin and gills of stressed animals was higher than that observed in controls (Table 2). In other tissues, no differences were detectable between control and stressed animals (Table 2).

Immunopositivity to 8-OHdG antibody was detected in the nuclei of: (i) epithelial cells of skin; (ii) epithelial cells lining the gastric mucosa; (iii) epithelial cells lining both filaments and lamellae of gills (Figure 5E, insert in (E)); (ii) enterocytes of the intestinal mucosa (Figure 5F); (iii) hepatocytes of liver parenchyma. No differences in terms of immunoreactivity were detected between controls and stressed animals (Table 2).

## 4. Discussion

Live fish transport is considered an essential procedure in aquaculture, but often exposes fish to stressors such as air exposure, handling, crowding and confinement [3,24,25]. Furthermore, it is known to cause the deterioration of transport-water conditions, reducing dissolved oxygen levels and pH and increasing ammonia nitrogen concentrations [26,27]. In fact, fish swimming activity during transportation period may lead to an increase in both respiration rate and nitrogenous waste [28], resulting in dissolved oxygen consumption and carbon dioxide as well as ammonia excretion in the water.

Fish react to stress by raising the levels of catecholamines and glucocorticoids such as cortisol hormone which is considered a primary stress indicator [29]. Increases in cortisol levels due to transport stress have already been described in both freshwater and marine species such as channel catfish (*Ictalurus punctatus*) [26,27] and Pacific bluefin tuna (*Thunnus orientalis*) [30]. Differently, in the present study, muscle cortisol levels significantly rose immediately after fish were loaded on the truck, indicating an increased stress level during the loading practice prior to the transport itself. In fact, after 16 h from departure, fish started to gradually recover, as attested by a significant decrease in cortisol level, but did not return to the basal values. A similar cortisol trend has been recently found by Wu et al. [31] in the tiger grouper serum (*Epinephelus fuscoguttatus*) during a simulated transport. This result possibly indicates that if, on one hand, meagre are able to rapidly face stress conditions associated with transport, on the other, it suggests that this stress response could be primarily associated to animals’ condition at departure, in terms of manipulation, net confinement and air exposure, or to the transport operator’s skills during loading, rather than to the transport itself.

It is worth noting that the cortisol trend observed in this study is well reflected by the molecular results. In fact, Real-Time PCR analysis evidenced that the highest expression of glucocorticoid receptor gene was detected in muscle of animals just after loading on the truck, whereas a decreasing trend of expression was observed during and after transport, returning to control levels immediately after transfer. These results confirm that the strong affinity between cortisol and the intracellular glucocorticoid receptor led to an increase in the GR mRNA levels [32,33].

Iwama et al. [34] reported that at the cellular level, the response of fish to stressors involves an increased synthesis of heat shock proteins. In particular, HSP70 expression has been observed in several fish species after exposure to various stress conditions [19,20,27,35,36,37,38,39,40,41,42,43,44,45,46,47,48,49]. In the present work, immunoreactivity to HSP70 antibody was detected in several tissues and organs although no differences were found between control and stressed animals. In a previous study, an increased expression of inducible HSP70 mRNA in larvae and fry of sea bass (*Dicentrarchus labrax*) subjected to transport stress as well as in skin and muscle of adults exposed to the same stress condition was demonstrated [19]. In the same work, Poltronieri et al. [19] evidenced that the presence of HSP70 protein was detectable only in skeletal muscle of transported animals. The immunohistochemical localization of HSP70 protein has been also demonstrated in the epithelia of renal tubules, gills and skin of carp (*Cyprinus carpio*) subjected to transport stress, whereas in trout (*Oncorhynchus mykiss*), HSP70 protein has been immunohistochemically detected only in red skeletal muscle and epidermis of control animals [20]. Recently, Refaey and Li [27] demonstrated that transported catfish exhibited a significantly higher level in the mRNA expressions of both hepatic HSP70 and HSP90 suggesting that the increment in HSPs may reflect the improved ability of channel catfish to adapt to transport stress.

During oxidative stress conditions, lipid peroxidation, protein carbonyl formation and DNA damage occur because more ROS are generated than cells can remove [12]. Lipid peroxides are unstable indicators of oxidative stress in cells that decompose to form more complex and reactive compounds such as 4-hydroxynonenal (HNE), which represents a natural by-product of lipid peroxidation [16]. In the experimental conditions used in this study, the anti-HNE antibody revealed positive immunostaining in the oropharyngeal cavity, stomach, liver, intestine, and skin, although no differences were detectable between control and transported animals. In a previous study, immunohistochemistry was used to investigate HNE cellular localization, showing that immunopositivity was mainly localized in melanomacrophage centers of kidney, liver, spleen, and ovary of the grass goby *Zosterisessor ophiocephalus* sampled in different areas of the Venice Lagoon, whereas animals from the detoxified control group did not exhibit any immunopositivity [23]. Recently, Fiocchi et al. [50] detected an increased HNE immunopositivity in several tissues of sea bass (*Dicentrarchus labrax*) subjected to handling and temperature increase.

In this work, nitrotyrosine (NT) as a marker for peroxynitrite production has been investigated by immunohistochemistry revealing the presence of the protein in various tissues of both transported and control animals. Moreover, the immunostaining intensity observed in skin and gills of transported animals was higher than that observed in controls. It is worth noting that increasing ammonia levels is considered one of the main fish stress inducers [28,51]. Therefore, the higher NT immunostaining intensity observed in this study could also be ascribed to the worsening of water quality parameters, as already reported in gilthead seabream juveniles exposed to ammonia changes [49]. In the grass goby *Zosterisessor ophiocephalus* sampled in different areas of the Venice Lagoon, the immunohistochemical localization of NT was detectable in melanomacrophage centers of spleen, kidney, liver and ovary, whereas animals from the detoxified control group did not exhibit any immunopositivity [23]. An increased NT immunopositivity has been recently observed in several tissues of sea bass (*Dicentrarchus labrax*) subjected to handling and temperature if compared to control animals [50]. 

An example of DNA damage due to bases’ modification is represented by the oxidation of deoxyguanosine to form 8-hydroxy-2’-deoxyguanosine (8-OHdG) [17]. In the present work, immunopositivity to 8-OHdG antibody was detected in the nuclei of several tissues although no differences in term of immunoreactivity was detected between controls and stressed animals. In a previous work, 8-OHdG expression and localization was investigated in the grass goby *Z. ophiocephalus*, collected in two sites of the Venice lagoon (Porto Marghera and Caroman) with different levels of pollution [52]. In particular, authors observed that in liver of males from Porto Marghera, the most polluted area, 8-OHdG expression was significantly greater than that observed in females, whereas animals from Caroman showed no sex-related differences. Changes in 8-OHdG activity as a product of oxidative DNA damage, histopathological changes and antioxidant responses have been recently investigated in rainbow trout exposed to linuron, a commonly used herbicide. Indeed, immunopositivity to 8-OHdG was detected both in liver and gill tissues of the treated animals, proving that an increase of tissue 8-OHdG, a well-known product of oxidative DNA damage, may be a consequence of oxidative stress [53].

## 5. Conclusions

In the present study, the teleost meagre *Argyrosomus regius* exposed to a 48 h transport clearly exhibited a stress response confirmed by marked changes in muscle cortisol and *gr* gene expression levels after loading, prior to the beginning of the transport. However, already after 16 h, a decreasing trend in the same stress indicators has been observed with a return of *gr* expression to the control levels still during transport, suggesting that this species is able to adapt rather quickly to these stress conditions. A strong NT immunopositivity was detected in epithelial cells of skin and gills of transported fish, likely due to deterioration in the water quality parameters during the transportation. Transport of live fish is an unavoidable practice in the aquaculture industry and thus it represents a critical welfare issue. Although research on welfare and morpho-physiological response to stressors of farmed fish species has begun to take hold in recent years, further studies are still needed in order to ensure better management of fish commercial transport, preserving animal health and welfare.

## Figures and Tables

**Figure 1 animals-11-01160-f001:**
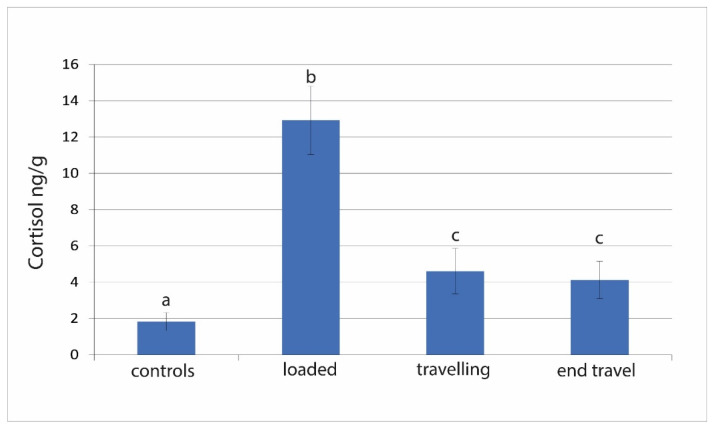
Cortisol concentrations detected in meagre muscle at four steps: at the fish farm, before transport; just loaded on the truck; during transport; at the end of transportation. Data are expressed as mean ± standard error (SE) (*n* = 20). a, b, c: Different letters denote statistically significant differences at the different sampling points (*p* < 0.0001).

**Figure 2 animals-11-01160-f002:**
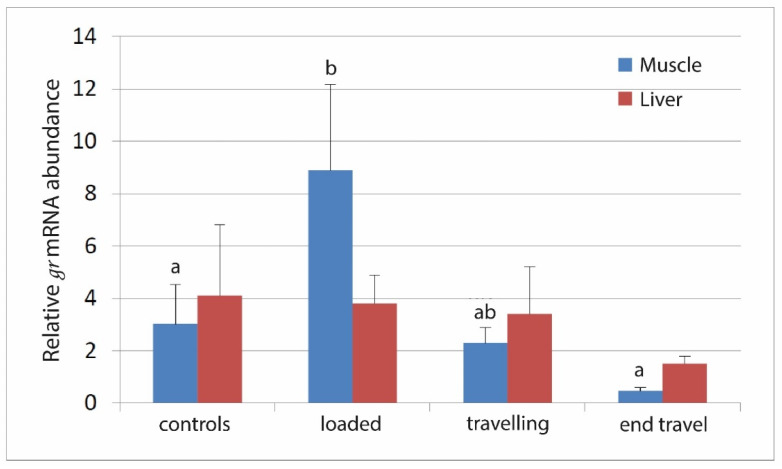
Relative expression of glucocorticoid receptor mRNA detected in meagre muscle and liver. Real-Time PCR levels were calculated at four steps: at the fish farm, before transport; just loaded on the truck; during transport; at the end of transportation. Values are means ± standard error (SE) (*n* = 10). a, b: Different letters denote statistically significant differences at the different sampling points, while the absence of letters indicates absence of significant differences.

**Figure 3 animals-11-01160-f003:**
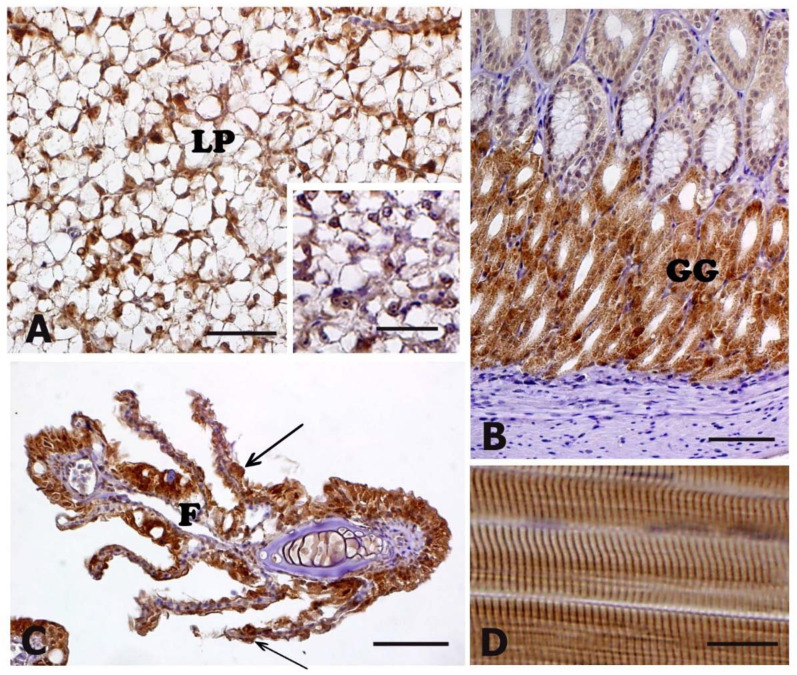
Heat shock protein 70 (HSP70) immunohistochemical localization in meagre (*Argyrosomus regius*). Panel (**A**) represents a control animal; insert in (**A**) and panels (**B**–**D**) represent animals exposed to transport stress. All panels are counterstained with Mayer’s haematoxylin. No differences in term of immunoreactivity were observed between control and transported animals (see also Table 2). (A) In liver, HSP70-immunostaining is present in the parenchyma (LP), at the level of the cytoplasm of hepatocytes. (B) In the stomach, immunopositivity is present in the gastric glands (GG). (C) The epithelial cells lining both filaments (F) and lamellae (arrows) show a marked HSP-70 immunoreactivity. (D) In skeletal lateral muscle, fibers exhibit a marked immunopositivity. Scale bars: (A–C) 20 μM; (D) 10 μM; insert in (A) 20 μM.

**Figure 4 animals-11-01160-f004:**
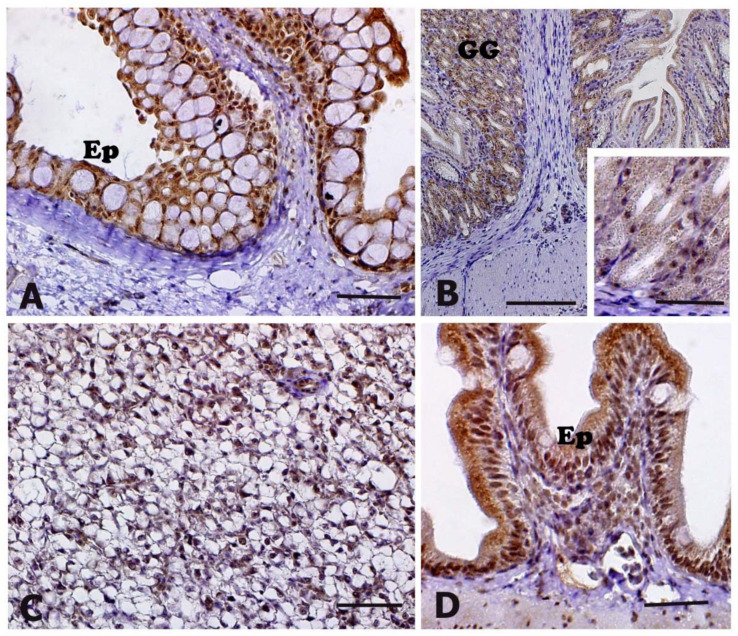
4-hydroxy-2-nonenal (HNE) immunohistochemical localization in meagre (*Argyrosomus regius*). Panels (**A)**, (**B**) and (**D**) represent animals exposed to transport stress, whereas panel (**C**) and the insert in (**B)** represent a control animal. No differences in term of immunoreactivity were observed between control and transported animals (see also Table 2). All panels are counterstained with Mayer’s haematoxylin. (**A**) The epithelium (Ep) of the oropharyngeal cavity exhibits a marked HNE-immunostaining. (**B**) In the stomach, a moderate HNE positivity is detectable in gastric glands (GG), although no differences in terms of reactivity were visible between control (insert in (**B**)) and stressed animals (**B**). (**C**) Liver of an animal from the control group, which exhibits immunostaining in the cytoplasm of hepatocytes. (**D**) In the intestine, a marked HNE immunopositivity was visible in the cytoplasm of enterocytes (Ep). Scale bars: (**A**–**D**) 20 μM, insert in (**A**) 20 μM.

**Figure 5 animals-11-01160-f005:**
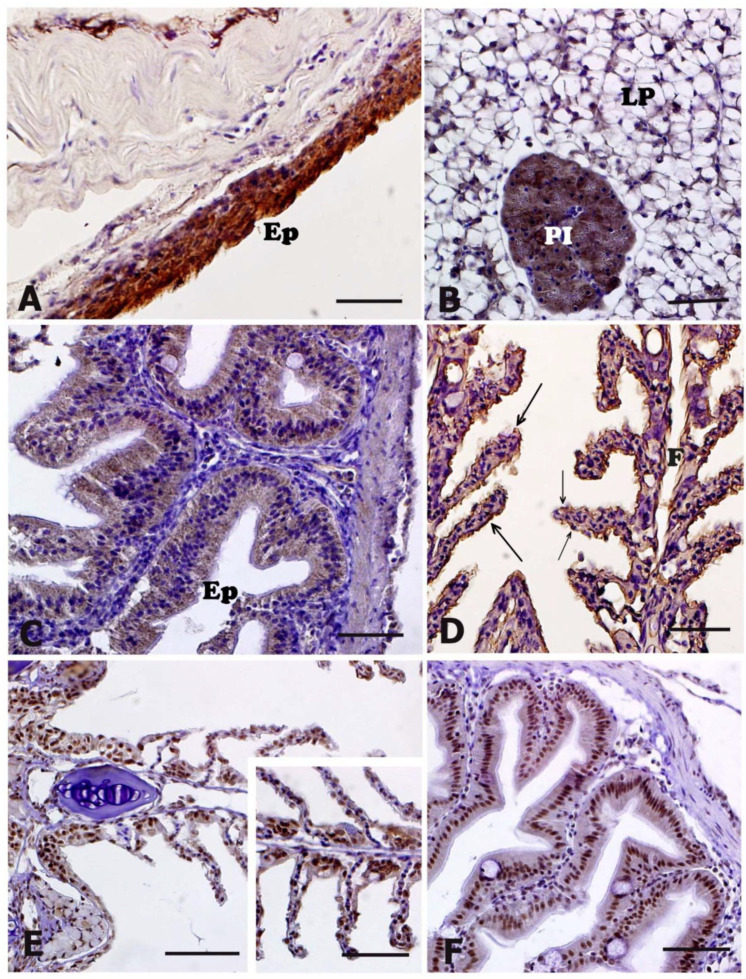
Nitrotyrosine (NT) in panels (**A–D**) and 8-hydroxy-2’-deoxyguanosine (8-OHdG) in panels (**E**,**F**) immunohistochemical localization in meagre (*Argyrosomus regius*). All panels are counterstained with Mayer’s haematoxylin and represent animals exposed to transport stress. Insert in (**E**) represents a control animal. (**A**) Skin of stressed animals which exhibits a marked NT immunopositivity in the epithelium (Ep). (**B**) Liver and pancreas of a stressed animal showing NT-immunostaining in both hepatocytes of the liver parenchyma (LP) and pancreatic islands (PI). (**C**) Intestine of a transported animal showing a moderate NT-immunostaining in the cytoplasm of enterocytes lining the epithelium (Ep). (**D**) Gills of a transported animal showing NT-immunostaining in the epithelial cells lining both filaments (**F**) and lamellae (arrows). (**E**) Gills of an animal from the stressed group showing 8-OHdG immunopositive nuclei in the epithelial cells lining both filaments and lamellae. Control animal (insert in (**E**)) shows a similar positivity. (**F**) 8-OHdG immunopositive nuclei in the epithelial cells lining the intestinal mucosa. Scale bars: (**A**–**F**) 20 μM, insert in (**E**) 20 μM.

**Table 1 animals-11-01160-t001:** Primary antibodies used.

Target Protein	Antibody	Species	Dilution	Characterization	Immunizing Antigen/Source
Heat Shock Protein 70 (HSP70)	Anti-HSP70	mouse monoclonal	1:600 (IHC)	Immunohistochemistry	Recombinant fragment from human Hsp70 (Abcam, UK)
4-hydroxy-2-nonenal (HNE)	Anti-HNE	mouse monoclonal	1:50 (IHC)	Immunohistochemistry	4-hydroxy-2-nonenal modified KLH (Abcam, UK)
Nitrotyrosine (NT)	Anti-NT	mouse monoclonal	1:1000 (IHC)	Immunohistochemistry	3-(4-hydroxy-3-nitrophenylacetamido) propionic acid conjugated to bovine serum albumin (BSA) (GeneTex, Inc., USA)
8 Hydroxy-2’-guanosine (8-OHdG)	Anti-8-OHdG	mouse monoclonal	1:3000 (IHC)	Immunohistochemistry	8-Hydroxy-2’-deoxyguanosine conjugated Keyhole Limpet Hemocyanin (Abcam, UK)

**Table 2 animals-11-01160-t002:** Immunohistochemical localization of heat shock protein 70 (HSP70), 4-hydroxy-2-nonenal (HNE), nitrotyrosine (NT) and 8-hydroxy-2’-deoxyguanosine (8-OHdG) in different tissues of meagre (transported and control animals): −, not detectable; +/−, slight but above background levels; +, moderate staining; ++, marked staining.

Tissue	CTRL				Transport Stress (End Travel)			
	HSP70	HNE	NT	8-OHdG	HSP70	HNE	NT	8-OHdG
Skin	+	+	+	+	+	+	++	+
Oropharyngeal cavity	+/−	++	+/−	+/−	+/−	++	+/−	+/−
Stomach	++	+	+/−	++	++	+	+	++
Intestine	+	++	+	+	+	++	+	+
Liver and pancreas	++	+	+	+	++	+	+	+
Gills	++	++	+	++	++	++	++	++
Muscle	++	+	+/−	−	++	+	+	+/−

## Data Availability

The data presented in this study are available within the article.

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
