# Peer review of "Muscle Cortisol Levels, Expression of Glucocorticoid Receptor and Oxidative Stress Markers in the Teleost Fish *Argyrosomus regius* Exposed to Transport Stress"

_animals, 2021, doi:10.3390/ani11041160_

Round 1
Reviewer 1 Report
The study entitled “Muscle cortisol levels, expression of glucocorticoid receptor and 2 oxidative stress markers.….” by Botoletti et al. reports interesting data regarding transport stress in meagre, fish of considerable importance in aquaculture. The manuscript is interesting and well written. It needs only some minor revisions:
Introduction
Page 2, line 63: modify “play an important role in….” with “play important roles in…”
Page 2 line 83: put abbreviation HSP70 after “Heat Shock Protein 70” and add a brief description;
Page 2, line 88: after “glucocorticoid receptor” add “in muscle and liver”
Materials and methods
Page 3 line 102: there is written “For each sampling point, 35 fish have been sacrificed”, but in successive paragraphs, other different numbers of sacrificed fish were reported for cortisol measurement, biomolecular analysis, and immunohistochemistry. Please, check carefully this point.
Page 3, line 108: please, delete “spleen” (no data in the text) and add “pancreas”
Results
In this section some terms must be modified:
Regarding fish gills, replace “primary and secondary” lamellae with the more appropriate terms “filaments” and “lamellae”.
The term “pharynx” should be replaced by “oropharyngeal” cavity or tract. The pharynx is not an independent tract of fish gut and it is mostly characterized by the presence of gills, which are considered as another sample.
The reactivity for HSP70 and HNE is in the basal part of stomach glands and not in gastric tips: please, modify legends fig 3b and 4b and the text.
The term “hepatopancreas” indicates the proper invertebrate organ. Please, replace this term with “liver” and “pancreas” in table 2, legends, and in the text.
Table 2 Please, specify “transport stress”: were sampling made in loaded animals, during or after traveling?
Figures:
Generally, in the figures arrows, arrowheads and asterisks have to be delated because they are misleading or useless. Often the immunoreactivity is not placed in single scattered cells, but quite diffused and easy to distinguish. Leave arrows only in fig 3C. Anyway, the Authors should add some abbreviations on images to explain anatomical structures ( for instance pancreas islet, filaments, lamellae….).
Reviewer 2 Report
In my opinion, the manuscript entitled “Muscle cortisol levels, expression of glucocorticoid receptor and oxidative stress markers in the teleost fish Argyrosomus regius exposed to transport stress” provide essential information in the field of aquaculture. The scientific background is sufficient, but there are some concerns about materials and methods, which are listed below.
The experiments are well designed and performed. The paper is well written.
The introduction is ok and properly reviewed the relevant literature.
Materials and methods:
Line 95: It is better to mention here that the fish was transferred from the fish farm.
It is not clear whether the transfer of fish was based on the general protocols used by fish farmers. For example, it is not clear whether the feeding of the fish was terminated before the transfer.
It seems that no sedatives or anaesthetics have been used to transfer the fish. Is this true?
Line 99: It should be noted here what is the ideal stocking density for the transfer of this fish species by the commercial truck and why did the authors consider this stocking density?
Line 99-100: As the authors point out in the discussion section of the article, changes in water quality during transportation are very important and can have a direct impact on the stress response of fish. Due to this issue, it is not clear why the physical and chemical parameters of water, especially nitrogenous compounds such as water ammonia, were not measured and presented.
The results presented are adequate.
The tables and figures used to show them are adequate.
The discussion is comprehensive.
Line 320: a natural by-product
The conclusion was supported by the results and clearly expressed the main hypothesis of the study.
Reviewer 3 Report
It is a well-known fact that stress is caused during loading (due to the loading protocol used) as well as during transport from deterioration of water quality parameters, further causing a substantial mortality rate in the transported fish (a number not mentioned in the manuscript). Statements in the manuscript are contradicted by results. There is no mention of water change in the tanks during transport (a common practice during transport of live fish especially during such a long trip). No data is provided regarding water quality in the tanks during transport (nitrates and nitrites) and no assessment was made for the time interval required for the transported juveniles to acclimate in their new environment. However, the manuscript offers an insight on physiological stress indicators in different time intervals of the transport as well as laboratory protocols and methods used for their assessment.
Here are several suggestions to improve the manuscript:
Ln 19: It should be stated that the fish assessed were actually juvenile and not mature.
Ln 29: Same as in 19.
Ln 23: the claim that fish recovered at the end of the transport is contradicted by Figure 1 which indicates that cortisol levels at the end of the transport have not returned to the control levels.
Ln 35: the claim that muscle gr expression was maintained 16 hours after departure, is contradicted by Figure 2 where it is indicated that during transport values are even lower compared to control levels.
Ln 100: Oxygen concentration levels 16-22 ppm seem very high; these are actually toxic levels that would cause additional stress; can you please recheck?
Ln 119: The transportation of juvenile fish of this size to the on growing (floating cage fish farm) site, is usually from a hatchery, not a fish farm.
Ln 182: …data were transformed…
Ln 201: Figure 2 indicates that relative gr mRNA abundance during transport, is lower compared to the controls, especially in muscle and seems that there should be a significant decrease compared to the loaded (non-significant difference is stated). Values are even lower at the end of the transport compared to the controls, a fact not discussed.
Ln 207: Non-parametric statistical tests used are not mentioned in the M&M section. Furthermore, were there used due to failed assumptions of ANOVA? Please elaborate.
Ln 366: gr mRNA abundance did not return to control levels at the end of transport, but during transport.
Author Response
Here are several suggestions to improve the manuscript:
Point 1: Ln 19: It should be stated that the fish assessed were actually juvenile and not mature.
Response 1: Authors stated that fish were juveniles. It has been clarified also in the Material and Methods section.
Point 2: Ln 29: Same as in 19.
Response 2: Authors modified the line according to the Reviewer’s clarification.
Point 3: Ln 23: the claim that fish recovered at the end of the transport is contradicted by Figure 1 which indicates that cortisol levels at the end of the transport have not returned to the control levels.
Response 3: Authors thank the Reviewer for the careful analysis and corrected the claim.
Point 4: Ln 35: the claim that muscle gr expression was maintained 16 hours after departure, is contradicted by Figure 2 where it is indicated that during transport values are even lower compared to control levels.
Response 4: Authors corrected the claim according to the Reviewer’s clarification.
Point 5: Ln 100: Oxygen concentration levels 16-22 ppm seem very high; these are actually toxic levels that would cause additional stress; can you please recheck?
Response 5: Yes, we confirm oxygen levels were those reported in the text. Usually, the transport personnel use higher than normal oxygen levels because the fish are stressed by the loading procedure.
Point 6: Ln 119: The transportation of juvenile fish of this size to the on growing (floating cage fish farm) site, is usually from a hatchery, not a fish farm.
Response 6: It was clarified in the text that it was a fish hatchery, as carefully noted by Reviewer.
Point 7: Ln 182: …data were transformed…
Response 7: The sentence has been corrected.
Point 8: Ln 201: Figure 2 indicates that relative gr mRNA abundance during transport, is lower compared to the controls, especially in muscle and seems that there should be a significant decrease compared to the loaded (non-significant difference is stated). Values are even lower at the end of the transport compared to the controls, a fact not discussed.
Response 8: Real Time PCR data were analyzed using the GLM (General Linear Model) approach, followed by a post-hoc comparison of HSD for unbalanced designs, which did not give significant difference between gr mRNA abundance in the “controls, travelling and end travel” animals, so we did not find it necessary to discuss the fact.
Point 9: Ln 207: Non-parametric statistical tests used are not mentioned in the M&M section. Furthermore, were there used due to failed assumptions of ANOVA? Please elaborate.
Response 9: We apologize for the mistake, this is a typographical error. We confirm that parametric tests have been used and the figure legend has been corrected.
Point 10: Ln 366: gr mRNA abundance did not return to control levels at the end of transport, but during transport.
Response 10: The sentence has been changed according to the Reviewer’s comment.
Reviewer 4 Report
This manuscript is a study about the effects of transport stress on the welfare status and stress response of Argyrosomus regius. The cortisol levels, glucocorticoid receptor (gr) mRNA expression and various oxidative stress markers were investigated after a 48-hour transport stress. This manuscript provides basic information about the stress status of fish during transportation.
However, I have some major concerns about this study are as follows.
- At the end of the abstract, the main conclusion and scientific significance of this study should to be succinctly described.
- There are too many keywords, and experimental techniques should not be used as keywords.
- It is necessary to have a space between the number and the unit except for temperature.
- For studies in fish, experimental fish are generally sampled randomly from at least three units, which were set up as biological duplications. As for this study, all the samples were selected only from one truck, which is insufficient. The experimental samples should be randomly taken from at at least 3 trucks.
- As glucocorticoid, cortisol is mainly synthesised in the interrenal tissue of kidney, and released into the bloodstream in response to stress. Therefore, the authors should have examined cortisol levels in the kidney and blood, but not the muscle.
- Four stages were selected during transportation, which was control (at the fish farm before transport), loaded (just loaded on the truck), travelling (during transport) and end travel (at the end of transportation). The water parameters for every stage, such as temperature, dissolved oxygen, and ammonia nitrogen, should be provided, rather than just providing the extreme values.
- Why did the authors measure receptor expression only in the liver and muscle, but not in other tissues? The relevant background information should be provided in the introduction section.
- As for immunohistochemistry, the results and data were poorly organized. How are the results in the Table 2 derived from? Which transport stage does the transport stress correspond to in the table and figures?
- As for Figure 3-5, the representative pictures in different tissues for each oxidative stress marker in the control and treatment groups should be provided. Actually, the results about the oxidative stress marker in this manuscript is tissue distribution pattern, but not their responses to transport stress.
Author Response
However, I have some major concerns about this study are as follows.
Point 1: At the end of the abstract, the main conclusion and scientific significance of this study should be succinctly described.
Response 1: A sentence has been added at the end of the abstract.
Point 2: There are too many keywords, and experimental techniques should not be used as keywords.
Response 2: The numbers of key words has been reduced, removing the experimental techniques.
Point 3: It is necessary to have a space between the number and the unit except for temperature.
Response 3: The space between number and unit has been added.
Point 4: For studies in fish, experimental fish are generally sampled randomly from at least three units, which were set up as biological duplications. As for this study, all the samples were selected only from one truck, which is insufficient. The experimental samples should be randomly taken from at least 3 trucks.
Response 4: To the best of our knowledge, in commercial conditions, only one truck has always been used. Specifically, different tanks are used in the same truck since the tanks are considered replicates of the same experimental design (truck). Please see the following references: Manuel et al. Stress in African catfish (Clarias gariepinus) following overland transportation. Fish Physiol Biochem, 40, 33–44 (2014). https://doi.org/10.1007/s10695-013-9821-7; Boerrigter et al. Recovery from transportation by road of farmed European eel (Anguilla anguilla). Aquac Res, 46, 1248-1260 (2015). https://doi.org/10.1111/are.12284; Refaey and Li. Transport stress changes blood biochemistry, antioxidant defense system, and hepatic HSPs mRNA expressions of channel catfish Ictalurus punctatus. Front Physiol, 9, 1–11, (2018). https://doi.org/10.3389/fphys.2018.01628
Point 5: As glucocorticoid, cortisol is mainly synthesised in the interrenal tissue of kidney and released into the bloodstream in response to stress. Therefore, the authors should have examined cortisol levels in the kidney and blood, but not the muscle.
Response 5: The muscle was selected given the unfavorable experimental (truck traveling) and working (blood sampling and centrifugation) conditions. Either way, it has already been validated as matrix for cortisol concentration measurements in a previously published paper. Furthermore, given the correlation between plasma and muscle cortisol in different species, muscle cortisol rises in a short time giving indications of acute stress. Please see Bertotto et al. Alternative matrices for cortisol measurement in fish. Aquac Res, 41, 1261-1267, (2010). https://doi.org/10.1111/j.1365-2109.2009.02417.x
Point 6: Four stages were selected during transportation, which was control (at the fish farm before transport), loaded (just loaded on the truck), travelling (during transport) and end travel (at the end of transportation). The water parameters for every stage, such as temperature, dissolved oxygen, and ammonia nitrogen, should be provided, rather than just providing the extreme values.
Response 6: It seemed more appropriate to report water temperature and dissolved oxygen levels as a range instead of single values for every stage since there were no notable variations. Moreover, commercial transporters also measured water ammonia nitrogen levels, but no record was made because the values were always below the acceptable threshold level. Furthermore, there are no regulations that indicate it as a mandatory procedure for commercial transports.
Point 7: Why did the authors measure receptor expression only in the liver and muscle, but not in other tissues? The relevant background information should be provided in the introduction section.
Response 7: Considering that this receptor is ubiquitously expressed, among the various tissues and organs that can be chosen, those of choice are liver and muscle as they are directly and strongly involved in gluconeogenesis (liver is the main site of gluconeogenesis) and protein catabolism (glucocorticoids, as cortisol, are known to induce a catabolic response in skeletal muscle). Background information with annex literature have been added in the revised manuscript. Please see: Benítez-Dorta et al. Total substitution of fish oil by vegetable oils in Senegalese sole (Solea senegalensis) diets: effects on fish performance, biochemical composition, and expression of some glucocorticoid receptor-related genes. Fish Physiol Biochem 39, 335–349 (2013). https://doi.org/10.1007/s10695-012-9703-4; Kuo et al. Metabolic functions of glucocorticoid receptor in skeletal muscle. Mol Cell Endocrinol, 380, 79-88, (2013). https://doi.org/10.1016/j.mce.2013.03.003; Sadoul and Vijayan, Stress and Growth. In Biology of Stress in Fish, eds C. B. Schreck, L. Tort, A. P. Farrell, and C. J. Brauner (London: Elsevier), 35, 167–205 (2016). https://doi.org/10.1016/B978-0-12-802728-8.00005-9; Palstra et al. Cortisol Acting Through the Glucocorticoid Receptor Is Not Involved in Exercise-Enhanced Growth, But Does Affect the White Skeletal Muscle Transcriptome in Zebrafish (Danio rerio). Front Physiol, 9, 1-11, (2019). https://doi.org/10.3389/fphys.2018.01889.
Point 8: As for immunohistochemistry, the results and data were poorly organized. How are the results in the Table 2 derived from? Which transport stage does the transport stress correspond to in the table and figures?
Response 8: Results reported in Table 2 and figures refer to the “end travel” stage and it has been clarified.
Point 9: As for Figure 3-5, the representative pictures in different tissues for each oxidative stress marker in the control and treatment groups should be provided. Actually, the results about the oxidative stress marker in this manuscript is tissue distribution pattern, but not their responses to transport stress.
Response 9: Authors thank the Reviewer for the observation. The results concerning immunohistochemistry are indicated in detail in Table 2. The semi-quantitative analysis did not reveal differences between control and transported animals. For this reason, the main purpose of the figures is to show in which organs/tissues and cell types the immunopositivity is located, more than to compare controls and transported animals.
Round 2
Reviewer 2 Report
-
Reviewer 4 Report
Although the manuscript has been better revised, however, in response to my previous review, the authors did not supplement relevant experimental data, and the explanation is not convincing. The major problems are as follows.
- For studies in fish, the number of experimental fish selected from one truck is the technical repetitions, but not biological duplications.
- Cortisol is mainly synthesized in the interrenal tissue of kidney, and released into the blood. Therefore, testing for cortisol in the blood is essential. Cortisol levels should be measured primarily in the blood and supplemented by other tissues. As indicated in the reference that provided by the authors, cortisol levels in the blood are six times higher than that in muscle. In fact, the paper examined cortisol levels not only in muscles, but also in the blood, mucous and intestines.
- The changes of oxidative stress markers during transportation should be detected. Unfortunately, the data about HSP70, HNE, NT and 8-OHdG in responses to transport stress were not provided.